# Increasing adverse drug reaction reporting— How can we do better?

**Miri Potlog Shchory**[1,2]☯*, **Lee H. Goldstein**[3], **Lidia Arcavi**[4], **Renata Shihmanter**[4], **Matitiahu Berkovitch**[5‡], **Amalia Levy**[1‡]

**1** Department of Public Health, Faculty of Health Sciences, Ben-Gurion University of the Negev, Beer Sheva, Israel, **2** Assaf Harofeh School of Nursing, Faculty of Medicine, The Hebrew University of Jerusalem, Jerusalem, Israel, **3** Clinical Pharmacology Unit, Haemek Medical Center, Afula affiliated to The Bruce Rapapport School of Medicine, Technion, Haifa, Israel, **4** Clinical Pharmacology Unit, Kaplan Medical Center, Hebrew University and Hadassah Medical School, Rehovot, Jerusalem, Israel, **5** Clinical Pharmacology Unit, Shamir Medical Center (Assaf Harofeh), Zerifin, Affiliated to Sackler School of Medicine, Tel-Aviv University, Tel-Aviv, Israel

☯ These authors contributed equally to this work.
‡ These authors also contributed equally to this work.
* miripo@shamir.gov.il

**Data Availability Statement:** All data files are available from the ICPSR database - project ID-openicpsr-119582.

**Funding:** The author(s) received no specific funding for this work.

## Abstract

Adverse drug reactions (ADRs) are associated with morbidity and mortality worldwide. Although national systems for reporting ADRs exist there is a low reporting rate. The aim of the current study was to evaluate an intervention plan for improving ADRs reporting among medical professionals (physicians and nurses). A multicentre intervention study was conducted, in which one medical centre was randomly assigned to the intervention group and two medical centres to the control group. The study consisted of 3 phases: baseline data collection, intervention and follow-up of the reporting rate. The questionnaire that was filled in at base line and at the end of study, contained questions about personal/professional demographic variables, and statements regarding knowledge of and behaviour toward ADRs reporting. The intervention program consisted of posters, lectures, distant electronic learning and reminders. An increase in the number of ADRs reports was noted in the intervention group (74 times higher than in the control group) during the intervention period, which was gradually decreased with as the study progressed (adjusted O.R = 74.1, 95% CI = 21.11–260.1, p<0.001). The changes in the "knowledge related to behaviour" (p = 0.01) and in the "behaviour related to reporting" (p<0.001) score was significantly higher in the intervention group. Specialist physicians and nurses (p<0.001), fulfilling additional positions (p<0.001) and those working in other places (p = 0.05) demonstrated a high rate of report. Lectures were preferable as a method to encourage ADRs reporting. The most convenient reporting tools were telephone and online reporting. Thus, implementation and maintenance of a continuous intervention program, by a pharmacovigilance specialist staff member, will improve ADRs reporting rates.

**Competing interests:** The authors have declared that no competing interests exist.

## Introduction

An integral part of drug therapy is the adverse reactions associated with the drug [1]. These reactions can cause personal injury, hospitalization overload and increase in health costs, thereby creating a heavy burden on national healthcare systems [2–5]. Studies around the world have demonstrated that 3–7% of all hospitalizations are a result of adverse drug reactions (ADRs) and 10–20% of inpatients suffer from drug-related adverse reactions [3, 6–8]. Serious ADRs were found to be the fourth to sixth causes of death in hospitalized patients in the US, leading to extended hospitalization and doubling of the cost of treating these patients [9,10]. Therefore, both healthcare teams and patients share a common goal of early detection and prevention of ADRs.

Pharmacovigilance is defined by the WHO as the science and activities relating to the detection, assessment, understanding and prevention of adverse reactions or any other drug-related problem. National systems for reporting drugs' adverse reactions can be found in almost every country [10–12]. Spontaneous reporting of medical staff regarding the occurrence of adverse reactions is the major source for monitoring and investigating adverse reactions of marketed drugs. However, only 1 in 20 adverse reactions is actually detected and defined as a real side effect; this leads to the erroneous assumption that the incidence of adverse reactions is much lower than it actually is [13–15]. Inadequate ADR reporting may lead to loss of clinical information that could prevent substantial damage to patients and consequently minimize health costs.

Hence, it is very important to encourage medical staff to report any definite or suspected ADR, as well as to establish and maintain an accurate database which can be used for analyzing and processing of accumulated data, drawing conclusions and providing further recommendations. This series of actions could optimize patients' wellbeing and safety and improve the functioning of the healthcare system.

Israel has been an official member of the World Health Organization international program for monitoring drugs since 1973. In August 2012, the Israeli ministry of health (MOH) published guidelines for reporting adverse events and new safety information. This document specifies the type of information that the Marketing Authorization Holder (MAH) requires reporting and enables the MAH to appoint a pharmacist to serve as a qualified person responsible for matters related to the reports included in this standard operating procedure, according to the standard worldwide practice. Those regulations were update at February 2013 in order to clarify the work processes related to reporting ADRs and new safety information, to update the definitions of the SOP and to update the types of information requiring reporting by the MAH. The aim of additional update from May, 2013 was to adjust the SOP to the Pharmacists Regulations [16]. A new regulation was launched at October, 2014, in which a reporting system in medical institutions for both common and severe ADRs was established. In July 2019, a new portal for reporting adverse events to the Risk Management Department was established. The purpose of all these regulations was to raise the awareness of the healthcare teams to the importance of reporting ADRs [17]. Nevertheless, the reporting rate in Israel, as is worldwide, is still quite low. Schwartzberg el al., identified 16,409 of Individual Case Safety Reports (ICSRs) submitted to the MOH's ADRs central database between September 2014 and August 2016. However, of these reports, only 5.5% were submitted by health care professionals from medical institutions, while 94.3% were submitted by pharmaceutical companies (MAH and importers) and only 0.2% of the reports have been submitted by patients and the general public [18].

The purpose of the present study was to establish and evaluate an intervention plan in order to improve the reporting norms of ADRs among medical professionals (physicians and nurses).

## Materials and methods

### Study design

The study was conducted over a period of 17 months (August 2012 to December 2013) and included the healthcare teams (physicians and nurses) of internal medicine divisions from three public medical centers in Israel: "A", "B" and "C", while each division served as a cluster. The medical centers selected for the study were public hospitals serving an urban and rural population of 0.5–1 million people each. In Israel, a high percentage of physicians and nurses are from the Commonwealth States and Russian is their first language. In order to make it easier for them to complete the questionnaire and to increase the response rate it was translated into Russian. Staff members who lacked sufficient knowledge of Hebrew or Russian to fill in the questionnaires were excluded from the study. Medical Center "A" was randomly selected to be the intervention group. Medical centers "B" and "C" were merged and served as the control group. The randomization among the centers was raffled by an external person who was not related to the study. ADRs reports were collected from the Israeli Ministry of Health computerized website for all three medical centers, and reports from medical center "A" (the intervention group) were also collected from the physical reporting binders available in the departments. The study was approved by the institutional review board of each medical center respectively. "A"—Ethics ("Helsinki") Committee; (protocol Number 180/10); "B"—Committee Helsinki; (protocol Number EMC-0107-10); "C"—Ethics ("Helsinki") Committee (protocol Number ver:1 KAP 1). In the introduction to the questionnaire a detailed explanation regarding the research and its rational was provided (See Supplement 1 in the S1 File). In this section the participant was required to give consent to be included in the research. Verbal consent was received during a meeting in which the research was presented to each participant. As part of the consent procedure the interviewer explained to each medical staff member (physicians and nurses) about the study. Potential participants were informed that taking part in the study was voluntary. Those who agreed to participate gave their oral consent. A record of all participants who provided oral consent was kept by the principal investigator. Every local IRB approved the use of verbal consent. Written consent was not obtained from the participants because the participants were staff members and not patients.

The study consisted of 3 phases: The first phase of baseline data collection lasted three months and included handing out a questionnaire to the healthcare teams. The questionnaire contained questions about personal and professional demographic data, and statements regarding the knowledge of and behavior toward ADRs reporting.

The second phase of the study was a 5-month intervention phase. The purpose of the intervention plan was to improve the staff reporting rate of ADRs. Site "A" was randomly selected as the medical center for intervention. The intervention program consisted of the following: a) posters for raising the awareness of the medical staff; these were placed at various locations throughout the departments, such as: physicians'/ nurses' rooms, medication rooms, and dining areas; b) Forty-five minute lectures that were given during divisional meetings of physicians and nurses separately. The lectures included: definitions of ADRs and pharmacovigilance, an explanation about the importance of the issue, data from international studies in the field, information about the relationship between adverse drug events and morbidity/mortality rates, incidence and prevalence of ADRs during hospital admission, the costs of ADRs for the healthcare system and the patients, presenting the reasons of ADR under reporting and a description of the current practices in Israel and around the world; c) Program promotion. This included: visiting the departments and talking with the medical staff twice weekly, presenting the program in the medical center portal and homepage and sending text messages to the participants on their mobile phones every two weeks (a total of 8 text messages were sent); d) introduction of distant electronic learning into the medical center portal.

The third phase of the study lasted 9 months and included following up on the monthly reporting rates. ADRs were reported through the Ministry of Health computerized website (for both the intervention and the control group) or documented in binders available only in the departments of the intervention group. At the end of this phase the participants from both groups were asked to fill in the same questionnaire again.

### The questionnaire (Supplement 1 in S1 File)

The questionnaire design was based on the causes for underreporting of ADRs among professionals, known as the "seven deadly sins" and on the combined theoretical model of the factors affecting the conditions for ADR reporting by healthcare professionals [17,19–32].

Demographic data included profession and degree, date and place of birth, gender, country of professional training, years of experience, expertise, as well as additional roles and positions in other medical institutions.

"Knowledge related to behavior" was intended to explore the knowledge of the participants about the ADR reporting procedure and its importance. It was investigated using the following general question which dealt with identification of ADRs and the reasons for reporting/not reporting: "You may notice an irregular adverse reaction from drug treatment and not report it since:" Then the participants were prompted to choose one of 5 statements: "A. I know the adverse drug reaction has already been documented by the pharmaceutical company"; "B. I do not know that there is a center for reporting adverse drug reactions"; "C. I am not aware of the need for reporting adverse drug reactions"; "D. I don't know how to report adverse drug reactions"; and "E. Reporting one adverse drug reaction does not significantly contribute to the reporting mechanism". The answers were ranged on a scale from 1 point—no reason for not reporting to 10 points—good reason for not reporting. This means that a staff member who reports ADRs will receive a lower score.

The score of "behavior related to reporting" was constructed from two items that analyzed the patterns of reporting to the National Center of pharmacovigilance and to pharmaceutical companies. The statements were: "A. I spoke with pharmaceutical companies about the possibility of adverse drug reactions with their drugs"; and "B. Have you ever reported adverse drug reactions to a national reporting center?". The average of this score ranged from 1 point—non reporting behavior—through 10 points—reporting behavior. The higher rate of reporting achieved a higher score.

The main hypothesis: after the intervention program there would be more ADRs reporting among medical professionals (physicians and nurses) in the intervention group compared to the control group and to ADRs reporting base line.

### Data analysis

Data analysis was carried out using the SPSS 21 software (PASW inc., USA). The statistical analysis was conducted according to the phases of the study. Means and standard deviation were used for continuous variables and examined by T or One-Way ANOVA/Mann Whitney tests based on the variables distribution. The score of "Knowledge related to behavior" and the score of "behavior related to reporting" didn't distribute normally, thus we used a- parametric test (Mann Whitney). Percentages and numbers were used for categorical variables and were tested by chi square or Fisher's exact tests as appropriate. Statistical significance was established as $p \leq 0.05$. The multi-variable models examined the independent effect of the intervention program on reporting. The difference (change) was viewed as the dependent variable, and factors forecasting change were examined through multi-variable linear regression models.

Identifying independent predicting factors of reporting adverse reactions was done through multi-variable logistic regression models.

Quantitative variables, such as the differences between groups, comparison between physicians and nurses and between medical centers, were analyzed by chi square or Fisher's exact tests. The differences and changes between the various parameters (knowledge and reporting patterns) were calculated by comparing the data collected after the intervention phase with the baseline data (change). The differences in knowledge and reporting among the study groups were compared separately by T tests with independent samples or through One-Way ANOVA. The building strategy for multivariable models was forced all the independent variables to one block. Both statistical and clinical justifications were considered. The models included all the variables that were found to be significant ($p < 0.05$) in the univariate data analysis, and covariates that were important to controlling for, as a baseline characteristic, according to the research questions. All the independent variables that were included in the analyses were presented in the results of the multivariable models. The multivariate models examined the effect of the intervention program on knowledge and reporting. The difference (change) was defined as the dependent variable and factors forecasting change were examined through linear regression. The adjusted factors of reporting adverse reactions (with 95% confidence intervals, 95% CI) were performed through multivariate logistic regression models. The independent effect of every measure index of the questionnaire was adjusted to demographic and professional variables.

## Results

433 (81.5%) medical staff members, physicians and nurses, completed the questionnaire twice, before and after the intervention. 47.8% of the participants were from the "A" medical center, 28.4% from "B" and 23.8% from "C". Distribution by gender was 69.1% females and 30.9% males. 73% were nurses and 27% physicians. No selection bias was found between the staff members completed the questionnaire the first time and those completed it twice. No differences in personal or professional variables were found between the intervention group ("A" medical center) and control group ("B" and "C" medical centers), except for the ratio between physicians and nurses and the subjects country of origin and average age."

During the research period, 336 ADRs were reported, of which 288 (85.7%) were reported in Medical Center "A", with 285 ADRs from the reporting binders and 3 ADRs from the Ministry of Health's computerized website. The ADRs reports from the control groups comprised 10 reports (3%) in center "B" and 38 reports (11.3%) in center C; these were reported to the Health Ministry's computerized website. The reports were checked and there were no duplicate reports by the staff members on both reporting channels in the intervention group.

The number of ADRs reports in the intervention vs. the control groups during the study period is presented in Fig 1.

A rapid and substantial increase in the number of ADR reports was noted in the intervention group during the 5-month intervention period, which gradually decreased toward the end of the study. The reporting in the intervention group went nearly back to baseline and was even lower than in the control group. On the other hand, there was almost no change in the number of ADR reports in the control group during the entire study duration. After the intervention period the reporting rate in the intervention group reverted to almost baseline and was lower than the control group, similarly to the trend that was observed in the baseline.

Comparison of the rate of ADRs reports received from physicians vs. nurses indicated that in both groups a substantial increase in the number of ADR reports was observed during the intervention period, which gradually decreased toward the end of the study.

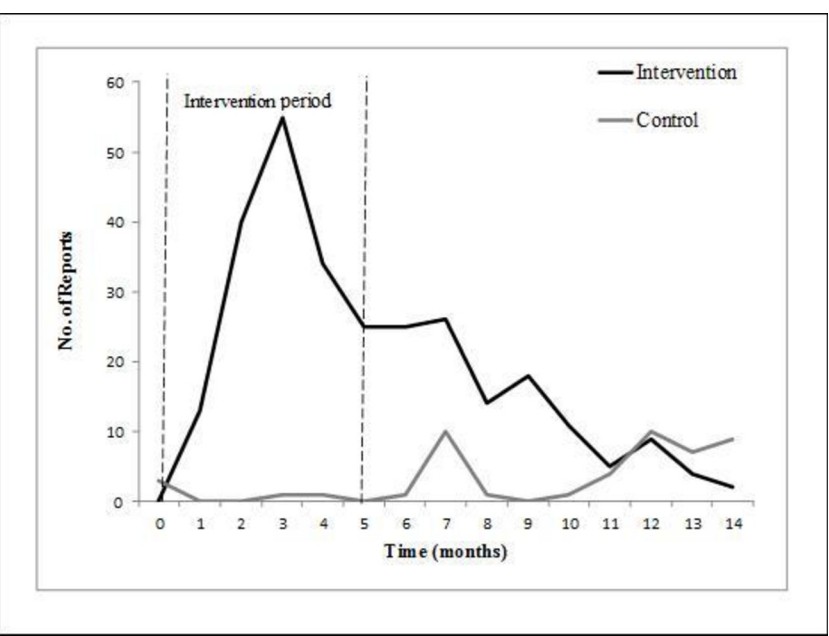

**Fig 1. Number of ADR reports in the intervention vs. the control groups during the study period.** A rapid and substantial increase in the number of ADR reports was noted in the intervention group during the 5-month intervention period, which gradually decreased toward the end of the study.

Comparison of the score of "knowledge related to behavior" showed that before any intervention the mean score of the control group was significantly lower than that observed in the intervention group (3.84±2.20 and 4.37±2.27 respectively, p = 0.02), demonstrating that the control group was more aware of ADRs reporting. Nevertheless, the changes in the "knowledge related to behavior" score was significantly higher in the intervention than in the control group (a change of -0.69±2.58 in the intervention group vs. -0.11±2.19 in the control group, P = 0.01)) Table 1).

When the score of "behavior related to reporting" was compared between the intervention and control groups upon intervention, a statistically significant increase in the score "behavior related to reporting" was observed in the intervention group, with a mean change of 0.65±2.22 (2.21±1.88 before intervention and 2.37±2.87 after intervention, P<0.001). No significant difference in the score of "behavior related to reporting" was noted in the control group (Table 2).

Comparison of patterns in "behavior related to reporting" was conducted according to various demographic and professional-related variables. The results revealed that the nurses demonstrated less changes in "behavior related to reporting" than physicians (P = 0.003). A significant positive correlation was found between the numbers of patients treated per day by the medical staff (nurses and physicians) and "behavior related to reporting", i.e. as the number

**Table 1. Comparison of the changes in the mean score of "knowledge related to behavior" of the intervention group vs. the control group.** The changes in the "knowledge related to behavior" score was significantly higher in the intervention than in the control group.

|  | Score before the intervention | Score after the intervention | Differnce in score changes |
| --- | --- | --- | --- |
|  | Mean ±SD (n) | Mean ± SD (n) | Mean ±SD(n) |
| **Intervention group** | 4.37±2.27 (206) | 3.67±2.16 (205) | - 0.69±2.58 (205) |
| **Control group** | 3.84±2.20 (220) | 3.73±2.14 (225) | - 0.11±2.19 (221) |
| **P value** | 0.02 | 0.79 | 0.01 |

Table 2. Comparison of the changes in the mean score of "behavior related to reporting" of the intervention group vs. the control group.

| | Score before the intervention | Score after the intervention | Differnce in scores changes |
|---|---|---|---|
| | Mean ± DS (n) | Mean ± DS (n) | Mean ± DS (n) |
| Intervention group | 2.21±1.88 (207) | 2.87±2.37 (207) | 0.65±2.22 (207) |
| Control group | 2.54±2.13 (226) | 2.48±2.12 (226) | - 0.06±2.16 (226) |
| p value | 0.09 | 0.79 | 0.001 |

of patients per caregiver increased, the change in "behavior related to reporting" score was higher (P = 0.02). In addition, the results revealed that the more aware the caregiver is of the fact that the patients are consuming more than one medication per day, a larger change in "behavior related to reporting" score is observed (P = 0.02, r = 0.13) (Table 3).

The demographic and the professional variables and reporting/non reporting behavior were later examined within the intervention group. As seen in Table 4, physicians reported more than nurses (56.9% vs. 36.5%, p = 0.009). Specialists, both nurses and physicians, reported more than non-specialists (60.6% vs. 32.6%, p<0.001). Those (both physicians and nurses) fulfilling additional positions and those working in other places beside the hospital demonstrated high rates of reports (66.7% vs. 33.6%, p<0.001 and 60.0% vs. 39.8%, p = 0.05, respectively) (Table 4).

Further analysis of the previous data to physicians and nurses revealed that the demographic and professional variables among the physicians did not have any effect on the percentage of ADRs reports. However, among the nurses, specialty (56.1% vs 29.6%, p = 0.02) and fulfilling additional positions (63.3% vs. 28.8%, p<0.0001) indeed increased reporting rates, while no difference was observed with regard to working in other places besides the hospital (37.5% vs. 36.6%, p = 0.96).

The intervention plan had a strong, independent and statistically significant effect on "behavior related to reporting" (p = 0.008). In addition, profession and number of patients treated per day by the caregiver also had a significant effect on the "behavior related to reporting" (p = 0.01 and p = 0.02, respectively) (Table 5).

In order to examine the independent effect of the intervention plan on reporting (yes / no), a logistic model was constructed. We found that the intervention plan had a strong, independent, statistically significant effect on the staffs' actual ADRs reporting. After standardization for specialist, expertise, holding managerial positions and those who work in other places, subjects in the intervention group reported 74 times higher than their counterparts in the control group (O.R = 74.1, 95% CI = 21.11–260.1, p<0.001) (Table 6).

Education and lectures were preferable, while payment for reporting was the least desirable method for encouraging medical staff to report side effects. The most convenient reporting tools were found to be the telephone and an internet site.

## Discussion

The purpose of this study was to establish and evaluate an intervention plan for increasing ADR reporting rate among physicians and nurses. This study demonstrates that the rate of ADR reporting increased significantly during the intervention period, and declined gradually thereafter. However, almost no change in the numbers of reports was observed in the control group during the entire duration of the study. The trend presented in the data that the rate of reports decreased over time after the intervention program was discontinued implies that continued intervention may be required to maintain the high rate of reports. Interestingly, the reporting rate in the control group was higher than in the intervention group at base line,

**Table 3. Comparison of patterns in "behavior related to reporting" according to various demographic- and professional-related variables of the intervention group vs. the control group.** Nurses demonstrated less changes in "behavior related to reporting" than physicians A significant positive correlation was found between the numbers of patients treated per day by the medical staff (nurses and physicians) and "behavior related to reporting", as well as between the awareness of the caregiver that the patients are consuming more than one medication per day, and the change in "behavior related to reporting" score.

| Variable | | Number | Change in "behavior related to reporting" (Mean±SD) | P-value |
|---|---|---|---|---|
| **Total No** | | | | |
| **Gender** | Female | 299 | 0.20±2.04 | P = 0.27 |
| 433 | Male | 134 | 0.46±2.57 | |
| **Profession** | Nurse | 316 | 0.06±1.91 | P = 0.003 |
| 433 | Physician | 117 | 0.89±2.79 | |
| **Country of origin** | Israel | 200 | 0.11±2.37 | P = 0.25 |
| 421 | Eastern Europe | 184 | 0.49±1.96 | |
| | Other | 37 | 0.38±2.52 | |
| **Country of professional training** | | | | |
| | Israel | 291 | 0.18±2.07 | P = 0.31 |
| 422 | Eastern Europe | 89 | 2.48±2.24 | |
| | Other | 42 | 0.62±3.05 | |
| **Specialty** | Yes | 144 | 0.42±2.79 | P = 0.40 |
| 433 | No | 289 | 0.21±1.93 | |
| **Fulfilling other positions** | Yes | 91 | 0.38±2.89 | P = 0.51 |
| 392 | No | 301 | 0.17±1.91 | |
| **Working in other places** | Yes | 47 | 0.70±3.01 | P = 0.29 |
| 423 | No | 376 | 0.22±2.11 | |
| **Age** | | 404 | r = 0.04 | P = 0.40 |
| 404 | | | | |
| **Age** | ≤35 | 143 | 0.17±2.21 | P = 0.70 |
| 404 | 36–45 | 131 | 0.32±2.23 | |
| | ≥46 | 130 | 0.40±2.41 | |
| **Years of seniority in the profession** | | 392 | r = 0.04 | P = 0.49 |
| 392 | | | | |
| **Years of seniority in the profession** | <7 | 110 | 0.17±2.21 | P = 0.48 |
| 392 | 7–20 | 172 | 0.18±2.13 | |
| | >21 | 110 | 0.50±2.38 | |
| **Years of seniority in the profession in Israel** | | 213 | r = 0.06 | P = 0.42 |
| 213 | | | | |
| **Years of seniority in the profession in Israel** | ≤20 | 113 | 0.45±1.76 | P = 0.72 |
| 213 | >21 | 96 | 0.35±2.21 | |
| **Years of Seniority in the internal division** | | 414 | r = 0.03 | p = 0.57 |
| 414 | | | | |
| **Years of Seniority in the internal division** | ≤3 | 137 | 0.16±1.97 | P = 0.31 |
| 414 | 3.1–13 | 140 | 2.55±2.22 | |
| | ≥13.1 | 137 | 0.23±2.43 | |
| **Number of patients treated per day** | | 382 | r = 0.12 | P = 0.02 |
| 382 | | | | |
| **Number of patients treated per day** | ≤12 | 195 | 0.06±2.11 | P = 0.02 |
| 382 | >13 | 187 | 1.56±2.34 | |
| **No. of drugs distributed/examined per day** | | 319 | r = -0.04 | P = 0.50 |
| 319 | | | | |
| **No. of drugs distributed/examined per day** | ≤50 | 190 | 2.42±0.41 | P = 0.71 |
| 319 | >51 | 129 | 2.09±0.31 | |
| **% of Patients taking more than one drug per day** | | 380 | r = 0.13 | P = 0.02 |

*(Continued)*

**Table 3.** (Continued)

| Variable | | Number | Change in "behavior related to reporting" (Mean±SD) | P-value |
|---|---|---|---|---|
| Total No | | | | |
| 380 | | | | |
| Patients taking more than one | <90% | 82 | 0.41±2.40 | P = 0.77 |
| drug per day | >90% | 298 | 0.38±2.34 | |
| 380 | | | | |

though it was not statistically significant. The intervention plan increased the reporting rate and the differences between the control and the intervention rose and were statistically significant throughout the intervention period. However, as distance from the intervention period increased, the amount of reports decreased eventually reverting to the trend that was observed at the baseline. The impact of the intervention waned with time. An example to this trend was demonstrated also in an educational intervention program to improve physicians' reporting of ADRs. In this study the reporting rate in the intervention group increased during the intervention, while it gradually decreased through 13 months of follow-up [9].

The compliance rate of the participants in the present study for filling in both questionnaires was rather high (81.5%). A lower rate of compliance was reported in similar studies, in which 22.8% (Biagi 2013) or 47% (Passier 2009) of the physicians answered questionnaires regarding ADRs reporting [33, 34]. Another study from Venezuela (Garciani 2011) reported higher compliance rate of 65.4% among physicians and pharmacists and 60% among nurses [35]. Interestingly, a much lower compliance rate was found in studies in which the questionnaires were sent to the participants by e-mail or regular mail [36]. The relatively high rate of compliance in the current study may be associated with the constant presence of the investigator at the medical centers and the personal contact with the study participants.

According to the results of the current study, physicians reported ADRs more than nurses. In addition, the change in the "Behavior related to reporting" score in the intervention group was higher among physicians compared to the nurses. The changes in the "Behavior related to reporting" is also demonstrated by the increase in the rate of ADRs in the intervention group compared the control group. This finding is consistent with other studies [1, 37–39]. In a study that took place in Korea, spontaneous reports of ADRs by e-mail were 13% among nurses vs. 53% reports by physicians. Some studies demonstrated that nurses mostly tend to report an ADR to a physician. Hanafi et al. have shown that 89% of the nurses who participated in the study said that they reported the ADR to the physician [40]. The Hajebi's study found that 56% of the nurses that come across a drug-related side effect reported to the department physician, 26% to the head nurse and 13% to a pharmacist [41]. The availability and accessibility of physicians to nurses who work in hospitals probably encouraged the reporting to physicians. This tendency could explain the difference in ADR reporting rates between nurses and physicians. Contrarily, in a research that was conducted in an Israeli public hospital where the medical staff was encouraged to report ADRs to the clinical pharmacology unit, the nurses were found to have reported more ADRs than the physicians [42]. In the present study, despite the fact that the nurses reported less than the physicians, their rate of reports peaked more quickly and decreased more slowly than that of the physicians.

We also found that professionals who fulfill additional positions in the department or in other health institutions (such as: community health services or treating senior citizens at nursing homes) demonstrated higher rates of ADR reporting. In addition, the results of the present study showed that specialists reported more ADRs than non-specialists and that the

**Table 4. Comparison of the demographic and the professional variables and reporting/non-reporting behavior in the intervention group vs. the control group.**
Physicians reported more than nurses; specialists, reported more than non-specialists; those fulfilling additional positions and those working in other places beside the hospital demonstrated high rates of reports.

| Variable (Total No.) | | Did not report | Reported | Total | p-value |
|---|---|---|---|---|---|
| | | % (n) | % (n) | | |
| **Gender** (207) | Female | 60.8 (96) | 39.2 (67) | 158 | 0.23 |
| | Male | 51.0 (25) | 49.0 (24) | 49 | |
| **Profession** (207) | Nurse | 63.5 (99) | 36.5 (57) | 156 | 0.009 |
| | Physician | 43.1 (22) | 56.9 (29) | 51 | |
| **Country of origin** (202) | Israel | 47.9 (34) | 52.1 (37) | 71 | 0.12 |
| | Eastern Europe | 63.4 (71) | 36.6 (41) | 112 | |
| | Other | 64.3 (11) | 42.1 (8) | 19 | |
| **Country of Professional Training** (200) | Israel | 55.9 (71) | 44.1 (56) | 127 | 0.83 |
| | Eastern Europe | 60.8 (31) | 39.2 (20) | 51 | |
| | Other | 59.1 (13) | 40.9 (9) | 22 | |
| **Specialty** (207) | Yes | 39.4 (26) | 60.6 (40) | 66 | <0.001 |
| | No | 67.4 (95) | 32.6 (46) | 141 | |
| **Fulfilling other positions** (194) | Yes | 33.3 (15) | 66.7 (30) | 45 | <0.001 |
| | No | 66.4 (99) | 33.6 (50) | 149 | |
| **Working in other places** (201) | Yes | 40.0 (10) | 60.0 (15) | 25 | 0.05 |
| | No | 60.2 (106) | 39.8 (70) | 176 | |
| **Age** | (Mean ±SD) | 57.13±7.21 | 42.87±9.34 | 103/82 | 0.92 |
| **Age** (185) | ≤35 | 48.8 (21) | 51.2 (22) | 43 | 0.08 |
| | 36–45 | 66.7 (44) | 33.3 (22) | 66 | |
| | ≥46 | 50.0 (38) | 50.0 (38) | 76 | |
| **Years of seniority in the profession** | (Mean ±SD) | 16.52±10.06 | 17.14±10.21 | 103/83 | 0.68 |
| **Years of seniority in the profession** (186) | <7 | 54.3 (19) | 45.7 (16) | 35 | 0.68 |
| | 7–20 | 58.6 (51) | 41.4 (36) | 87 | |
| | >21 | 51.6 (33) | 48.4 (31) | 64 | |
| **Years of seniority in the profession in Israel** | (Mean ±SD) | 19.64±9.05 | 21.4±10.83 | 81/47 | 0.32 |
| **Years of seniority in the profession in Israel** (128) | ≤20 | 64.8 (46) | 35.2 (25) | 71 | 0.69 |
| | >21 | 61.4 (35) | 38.6 (22) | 57 | |
| **Years of Seniority in the internal division** | (Mean ±SD) | 11.26±9.49 | 13.36±10.11 | 114/84 | 0.32 |
| **Years of Seniority in the internal division** (198) | ≤3 | 65.4 (34) | 34.6 (18) | 52 | |
| | 3.1–13 | 60.3 (41) | 39.7 (27) | 68 | 0.19 |
| | ≥13.1 | 50.0 (39) | 50.0 (39) | 78 | |
| **No. of patients treated per day** | (Mean ±SD) | 15.92±9.22 | 16.34±11.19 | 104/79 | 0.78 |
| **No. of patients treated per day** (183) | ≤12 | 54.5 (54) | 45.5 (45) | 99 | 0.50 |
| | ≥13 | 59.5 (50) | 40.5 (34) | 84 | |
| **No. of drugs distributed/examined per day** | (Mean ±SD) | 53.72±43.90 | 65.04±54.71 | 91/69 | 0.15 |
| **No. of drugs distributed/examined per day** (160) | ≤50 | 59.1 (55) | 40.9 (38) | 93 | 0.50 |
| | >51 | 53.7 (36) | 46.3 (31) | 67 | |
| **% of Patients taking more than one drug per day** | (Mean ±SD) | 89.44±19.70 | 87.19±24.35 | 102/77 | 0.56 |
| **Patients taking more than one drug per day** (179) | <90 | 58.3 (21) | 41.3 (15) | 36 | 0.86 |
| | >90 | 56.6 (81) | 43.4 (62) | 143 | |

rate of ADR reporting was associated with the number of patients treated per day by the caregiver. A contrary observation was reported by a study in Ireland, which investigated the rate of ADR reporting among 118 hospital-based physicians. This study found a higher rate of ADR reporting among general physicians than among surgeons [43].

**Table 5. The effect of the intervention program and demographic characteristics on "behavior related to reporting".** The intervention plan had a strong, independent and statistically significant effect on "behavior related to reporting". Profession and number of patients treated per day by the caregiver also had a significant effect on the "behavior related to reporting".

| Variable | β | p value |
|---|---|---|
| Intervention (Yes / No) | 0.13 | 0.008 |
| Professional (physician / nurse) | 0.14 | 0.01 |
| Number of patients treated per day | 0.12 | 0.02 |
| Gender (Male / Female) | 0.009 | 0.87 |

The present study demonstrated that the preferred method for increasing the rate of ADR reporting was lectures and education A study that compared telephone-interview intervention with a workshop intervention showed that the latter increased ADR reporting rate by a four-fold on average compared to the control group over 20 months post-intervention. However, no significant difference vs. the control group in ADR reporting was found in the telephone-interview intervention [44]. Other studies have shown that improved communication with fellow physicians and involvement of pharmacists might be the best ways to improve ADR reporting [21, 34]. Regular newsletter on current awareness in drug safety, information on new ADRs, and international drug safety information were also identified as tools or methods that may motivate ADR reporting in a study conducted by Santosh et al. among 450 healthcare professionals working at Regional Pharmacovigilance Centers in Nepal [21].

Our research shows that the preferred means of reporting were telephone or website. Other studies also report these methods as preferential. A study of 500 nurses from a teaching hospital in Teheran showed that among the 10% who reported an ADR, the majority of the nurses preferred using the telephone [45]. Among physicians in India and the Netherlands, most of the ADRs were reported using the computerized system [34, 46].

Payment for reporting was found to be the least favored method to encourage ADR reporting in the current study. A survey of 91 practice nurses, health visitors, school nurses and general physicians conducted by Pulford et al., has shown that payment for ADR reporting was indeed the least acceptable out of 14 other options of gratuity [47].

## Conclusions

The results of this study indicate that training and educating medical practitioners and providing them with relevant knowledge regarding ADR reporting is essential. Due to the observation that the reporting rate decreased with time upon the finalization of the intervention period, it seems that maintaining a program to encourage reporting, is necessary. Regular implementation of such a program in the healthcare system will increase the awareness of the medical staff and improve reporting rates. Maintaining the intervention program could be carried out by nomination of a pharmacovigilance specialist trustee to administer a routine intervention program. This expert could be a physician, a nurse or a pharmacist. Visits, personal

**Table 6. The independent effect (adjusted Odd Ratios and 95% CI) of the intervention program on ADR reports.** A logistic model revealed that the intervention plan had a strong, independent, statistically significant effect on the staffs' actual ADRs reporting.

| Variable | O.R | CI 95% | p value |
|---|---|---|---|
| Intervention (Yes / No) | 74.1 | 21.11–260.1 | 0.001> |
| Fulfilling other roles (Yes / No) | 3.16 | 1.45–6.92 | 0.004 |
| Specialty (Yes / No) | 2.32 | 1.14–4.70 | 0.02 |
| Professional (physician / nurse) | 2.70 | 1.31–5.57 | 0.007 |

discussions, posters, lectures about the importance of ADRs reporting and how to carry it out and sending text messages to the medical staff on their mobile phones with reminders and relevant information should be used in order to continuously raise their awareness reporting about ADRs. In addition, the personal contact of the medical staff with the trustee will encourage their commitment to report about ADRs. This will probably improve monitoring of medication use, decrease morbidity/mortality rates and hospitalization duration.

## Limitations

The study was carried out in internal divisions of only 3 public medical centers, and therefore may not have an external validation in other hospitals.

The study included only physicians and nurses, therefore the results cannot be applied to additional medical professionals.

There were some differences in the basic characteristics between the clusters (hospitals) which may have affected the quality of the intervention to a certain extent. However, after adjusting for the demographic variables, we can assume that the results of the study are indeed due to the effect of the intervention.

We cannot assume from this study that the effect of the intervention would improve safety of medicine use in the long term and reduce health costs.

## Supporting information

**S1 Data.**
(SAV)

**S1 File.**
(DOC)

**S2 File.**
(DOC)

## Author Contributions

**Conceptualization:** Miri Potlog Shchory, Lee H. Goldstein, Lidia Arcavi, Renata Shihmanter, Matitiahu Berkovitch, Amalia Levy.

**Data curation:** Miri Potlog Shchory, Matitiahu Berkovitch.

**Formal analysis:** Miri Potlog Shchory, Amalia Levy.

**Investigation:** Miri Potlog Shchory.

**Methodology:** Miri Potlog Shchory, Lee H. Goldstein, Lidia Arcavi, Renata Shihmanter, Matitiahu Berkovitch, Amalia Levy.

**Supervision:** Amalia Levy.

**Validation:** Miri Potlog Shchory.

**Writing – original draft:** Miri Potlog Shchory.

**Writing – review & editing:** Matitiahu Berkovitch, Amalia Levy.

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
