## [Decision Letter · Decision Letter 0]

28 Sep 2019

PONE-D-19-24600

Increasing Adverse Drug Reaction Reporting –

How can we do better?

PLOS ONE

Dear Dr. Shchory Potlog,

Thank you for submitting your manuscript to PLOS ONE. After careful consideration, we feel that it has merit but does not fully meet PLOS ONE’s publication criteria as it currently stands. Therefore, we invite you to submit a revised version of the manuscript that addresses the points raised during the review process.

We would appreciate receiving your revised manuscript by Nov 12 2019 11:59PM. To enhance the reproducibility of your results, we recommend that if applicable you deposit your laboratory protocols in protocols.io, where a protocol can be assigned its own identifier (DOI) such that it can be cited independently in the future. For instructions see: http://journals.plos.org/plosone/s/submission-guidelines#loc-laboratory-protocols

We look forward to receiving your revised manuscript.

Kind regards,

Karen Cohen

Academic Editor

PLOS ONE

Journal Requirements:

2. Thank you for including your ethics statement: The study was approved by the institutional review board of each medical center respectively.

4. Please provide additional details regarding participant consent. In the ethics statement in the Methods and online submission information, please ensure that you have specified (1) whether consent was informed and (2) what type you obtained (for instance, written or verbal, and if verbal, how it was documented and witnessed). If consent was implied due to the completion and returning of the questionnaire, please state this in the Methods section.

4. Please include your tables as part of your main manuscript and remove the individual files. Please note that supplementary tables (should remain/ be uploaded) as separate "supporting information" files

5. Please amend the manuscript submission data (via Edit Submission) to include authors Lee H. Goldstein, Lidia Arcavi, Renata Shihmanter, Matitiahu Berkovitch and Amalia Levy.

6. Please amend either the abstract on the online submission form (via Edit Submission) or the abstract in the manuscript so that they are identical.

7.  We note that you have indicated that data from this study are available upon request. PLOS only allows data to be available upon request if there are legal or ethical restrictions on sharing data publicly. For information on unacceptable data access restrictions, please see http://journals.plos.org/plosone/s/data-availability#loc-unacceptable-data-access-restrictions.

Reviewers' comments:

Reviewer's Responses to Questions

**Comments to the Author**

1. Is the manuscript technically sound, and do the data support the conclusions?

Reviewer #1: Partly

Reviewer #2: Yes

2. Has the statistical analysis been performed appropriately and rigorously? 

Reviewer #1: I Don't Know

Reviewer #2: Yes

3. Have the authors made all data underlying the findings in their manuscript fully available?

Reviewer #1: Yes

Reviewer #2: No

4. Is the manuscript presented in an intelligible fashion and written in standard English?

Reviewer #1: No

Reviewer #2: No

5. Review Comments to the Author

Reviewer #1: Dear Colleagues

Thank you for the opportunity to review and comment on your reserach. I am aware of the amount of work done and my comments aim at helping you improve your manuscript.

The first thing that suprises me is that your research dates from 2013 and 6 years have elapsed since then, some statements you make (see below) might be out of date.

The references used in the introduction are old/very old:

81-84: "...new legislation was recently launched...2014" or "the reporting rate in Israel...is still very low", this with a reference from 1999. While underreporting might still be a problem, I would be very surprised that nothing has changed in the past 20 yrs.Please update with actual recent figures.

Methods:

133-142: the answers to the question with "because" do not make sense in the context of the questions, while valid as stand alone. Please clarify.

To the reader it is not clear why the focus was on Hebrew (obvious) and Russian speakers (much less obvious). Please explain.

143-148: you mention two statements analyzed, but B is actually a question. Please clarify. Also, behaviour change cannot be measured by subjective statements. In this case behaviour change can be measured by the number of reports submitted before and after the intervention.

The statistical analysis of the results of the questionnaires is very detailed but unfortunately does not lead to conlusions on how to improve future interventions.

Results:

You write that the reporting in the intervention group was higher than in the control and that the effect diminished over time. What you do not mention is that according to Fig 1, the reporting in the intervention group went nearly back to baseline and was even lower than in the control group. While the single values might not reach statistical significance, the trend is still impressive and must be mentioned and discussed - not doing so misleads the reader. Please rectify (179-183).

Discussion:

see comment above

Conclusion:

see above and please explain more in detail how interventions can be improved based on your data and their analysis.

Language: the manuscript would benefit from a revision by a native English speaker.

Reviewer #2: Methods section: description (grade) is not very clear. Scoring and description seem to contradict. Should be rephrased.

Inconsistent/non standardized/interchangeable use of the terms side effects, adverse effects, adverse reactions.

Pharmacovigilance is described as a pharmacy practice. I am not aware of any such categorization.

How were duplicate reports identified/handled (since for Group A, post intervention reports came both from the facility and the MoH website.

Interventions such s these are known to have a temporary positive impact on reporting. What is more important is to discuss how to sustain this. Also, will such an intervention lead to quality improvements?

Authors have explained that there is no data repository. And that primary data could be provided as email attachments.

6. PLOS authors have the option to publish the peer review history of their article (what does this mean?). If published, this will include your full peer review and any attached files.

Reviewer #1: No

Reviewer #2: No

---

## [Author Response · Author response to Decision Letter 0]

16 Dec 2019

Reviewer #1

• "Thank you for the opportunity to review and comment on your research. I am aware of the amount of work done and my comments aim at helping you improve your manuscript. The first thing that surprises me is that your research dates from 2013 and 6 years have elapsed since then, some statements you make (see below) might be out of date.

The references used in the introduction are old/very old: 81-84: "...new legislation was recently launched...2014" or "the reporting rate in Israel...is still very low", this with a reference from 1999. While underreporting might still be a problem, I would be very surprised that nothing has changed in the past 20 yrs. Please update with actual recent figures."

Answer: Newer references were used in the revised manuscript, as recommended. This information was added to the "Introduction" (Page 5, Line 91) of the revised version and it now appears as follow: 

In August 2012, the Israeli ministry of health (MOH) published guidelines for reporting adverse events and new safety information. This document specifies the type of information that the Marketing Authorization Holder (MAH) requires reporting and enables the MAH to appoint a pharmacist to serve as a qualified person responsible for matters related to the reports included in this standard operating procedure, according to the standard worldwide practice. Those regulations were update at February 2013 in order to clarify the work processes related to reporting ADRs and new safety information, to update the definitions of the SOP and to update the types of information requiring reporting by the MAH. The aim of additional update from May, 2013 was to adjust the SOP to the Pharmacists Regulations (16). A new regulation was launched at October, 2014, in which a reporting system in medical institutions for both common and severe ADRs was established. In July 2019, a new portal for reporting adverse events to the Risk Management Department was established.

Data regarding the low reporting rate in Israel was updated - Page 5, line 107, ref.18: 

 Schwartzberg el al., identified 16,409 of Individual Case Safety Reports (ICSRs) submitted to the MOH’s ADRs central database between September 2014 and August 2016. However, of these reports, only 5.5% were submitted by health care professionals from medical institutions, while 94.3% were submitted by pharmaceutical companies (MAH and importers) and only 0.2% of the reports have been submitted by patients and the general public [18].

• "Methods: 

133-142: the answers to the question with "because" do not make sense in the context of the questions, while valid as stand alone. Please clarify."

Answer: This question refers to the reasons for not reporting after identifying adverse reaction effects of drug therapy. It contains 5 possible causes for not reporting. The word “because” has been replaced by the word “since”- Page 9, line 182. 

• "To the reader it is not clear why the focus was on Hebrew (obvious) and Russian speakers (much less obvious). Please explain."

Answer: In Israel, a high percentage of physicians and nurses are from the Commonwealth States and Russian is their first language. In order to make it easier for them to complete the questionnaire and to increase the response rate it was translated into Russian.

This explanation was added in the revised version, as suggested (Page 7, line 123).

• "143-148: you mention two statements analyzed, but B is actually a question. Please clarify."

Answer: The word “statements” has been changed to “items” – Page 11, line 191. 

• "Also, behavior change cannot be measured by subjective statements. In this case behavior change can be measured by the number of reports submitted before and after the intervention."

Answer: Indeed, as stated by the reviewer, behavior change cannot be measured by subjective statements. However, it could be measured by the number of reports submitted before and after the intervention. This is presented in the data of Figure 1, where a rapid and substantial increase in the number of ADR reports was noted in the intervention group during the intervention period while almost no change in the number of ADRs reports was observed in the control group. This information is presented in the "Discussion" of the revised manuscript, as recommended, and it now appears: (Page 19, line 346)

The trend presented in the data that the rate of reports decreased over time after the intervention program was discontinued implies that continued intervention may be required to maintain the high rate of reports. Interestingly, the reporting rate in the control group was higher than in the intervention group at base line, though it was not statistically significant. The intervention plan increased the reporting rate and the differences between the control and the intervention rose and were statistically significant throughout the intervention period. However, as distance from the intervention period increased, the amount of reports decreased eventually reverting to the trend that was observed at the baseline.

• "The statistical analysis of the results of the questionnaires is very detailed but unfortunately does not lead to conclusions on how to improve future interventions."

Answer: According to the statistical analyzed data, the increase in the rate of ADRs in the intervention group was noted during the 5-month intervention period. Nevertheless, it was decreased toward the end of the study. Therefore, it was suggested that the intervention program, as well as training and educating the medical practitioners should be maintained. This was addressed to the conclusion (page 22, line 424) 

Maintaining the intervention program could be carried out by nomination of a pharmacovigilance specialist trustee to administer a routine intervention program. This expert could be a physician, a nurse or a pharmacist. Visits, personal discussions, posters, lectures about the importance of ADRs reporting and how to carry it out and sending text messages to the medical staff on their mobile phones with reminders and relevant information should be used in order to continuously raise their awareness reporting about ADRs. In addition, the personal contact of the medical staff with the trustee will encourage their commitment to report about ADRs.

and Abstract (page 2, line 51).

Thus, implementation and maintenance of a continuous intervention program, by a pharmacovigilance specialist staff member, will improve ADRs reporting rates. 

• "Results: You write that the reporting in the intervention group was higher than in the control and that the effect diminished over time. What you do not mention is that according to Fig 1, the reporting in the intervention group went nearly back to baseline and was even lower than in the control group." 

Answer: Indeed as the reviewer commented, although the reporting in the intervention group was lower than that of the control group, it was not statically significant. The following sentence was added to the revised version, as suggested and it appears:

After the intervention period the reporting rate in the intervention group reverted to almost baseline and was lower than the control group, similarly to the trend that was observed in the baseline.

(Page 12, line 244)

• "While the single values might not reach statistical significance, the trend is still impressive and must be mentioned and discussed - not doing so misleads the reader. Please rectify (179-183)."

Answer: This information was added to the "Discussion" of the revised manuscript page 19, line 346.

The trend presented in the data that the rate of reports decreased over time after the intervention program was discontinued implies that continued intervention may be required to maintain the high rate of reports. Interestingly, the reporting rate in the control group was higher than in the intervention group at base line, though it was not statistically significant. The intervention plan increased the reporting rate and the differences between the control and the intervention rose and were statistically significant throughout the intervention period. However, as distance from the intervention period increased, the amount of reports decreased eventually reverting to the trend that was observed at the baseline.

• "Discussion:

see comment above" 

Answer: The discussion section was amended according to the comment – page 20, line 346

• "Conclusion:

see above and please explain more in detail how interventions can be improved based on your data and their analysis."

Answer: The conclusion section was amended according to the comment (page 22, line 424): 

Maintaining the intervention program could be carried out by nomination of a pharmacovigilance specialist trustee to administer a routine intervention program. This expert could be a physician, a nurse or a pharmacist. Visits, personal discussions, posters, lectures about the importance of ADRs reporting and how to carry it out and sending text messages to the medical staff on their mobile phones with reminders and relevant information should be used in order to continuously raise their awareness reporting about ADRs. In addition, the personal contact of the medical staff with the trustee will encourage their commitment to report about ADRs. 

• "Language: the manuscript would benefit from a revision by a native English speaker."

Answer: The manuscript was reviewed by a professional native English speaker.

Reviewer #2: 

• "Methods section: description (grade) is not very clear. Scoring and description seem to contradict. Should be rephrased."

Answer: Thank you for this very correct note. The word “grade” has been replaced by the word “score” all over the text and in the tables.

• "Inconsistent/non standardized/interchangeable use of the terms side effects, adverse effects, adverse reactions."

Answer: The terms side effects and adverse effects were changes to "adverse reactions" all over the text. 

• "Pharmacovigilance is described as a pharmacy practice. I am not aware of any such categorization."

Answer: Thank you for the remark. The definition of Pharmacovigilance by the WHO was introduced into the text of the revised version, as suggested– (Page 4, line 74): 

Pharmacovigilance is defined by the WHO as the science and activities relating to the detection, assessment, understanding and prevention of adverse reactions or any other drug-related problem

• "How were duplicate reports identified/handled (since for Group A, post intervention reports came both from the facility and the MOH website."

Answer: The reports were checked and there were no duplicate reports by staff members on both reporting channels in the intervention group. This information was added to the revised manuscript, as recommended (Page 11 Line 228)

The reports were checked and there were no duplicate reports by the staff members on both reporting channels in the intervention group.

• "Interventions such as these are known to have a temporary positive impact on reporting. What is more important is to discuss how to sustain this. Also, will such an intervention lead to quality improvements?"

Answer: This issue was already discussed –page 22, line 424

• "Authors have explained that there is no data repository. And that primary data could be provided as email attachments."

Answer: All needed data is presented.

---

## [Decision Letter · Decision Letter 1]

2 Mar 2020

PONE-D-19-24600R1

Increasing Adverse Drug Reaction Reporting - How can we do better?

PLOS ONE

Dear Dr. Shchory Potlog,

Thank you for submitting your manuscript to PLOS ONE. After careful consideration, we feel that it has merit but does not fully meet PLOS ONE’s publication criteria as it currently stands. Therefore, we invite you to submit a revised version of the manuscript that addresses the points raised during the review process.

 In particular two of the reviewers of the revised manuscript  raised a number of questions regarding  the statistical analysis, as well as some queries regarding study design. Please can you address the issues they have raised?

We would appreciate receiving your revised manuscript by Apr 16 2020 11:59PM. To enhance the reproducibility of your results, we recommend that if applicable you deposit your laboratory protocols in protocols.io, where a protocol can be assigned its own identifier (DOI) such that it can be cited independently in the future. For instructions see: http://journals.plos.org/plosone/s/submission-guidelines#loc-laboratory-protocols

We look forward to receiving your revised manuscript.

Kind regards,

Karen Cohen

Academic Editor

PLOS ONE

Reviewers' comments:

Reviewer's Responses to Questions

**Comments to the Author**

1. If the authors have adequately addressed your comments raised in a previous round of review and you feel that this manuscript is now acceptable for publication, you may indicate that here to bypass the “Comments to the Author” section, enter your conflict of interest statement in the “Confidential to Editor” section, and submit your "Accept" recommendation.

Reviewer #1: All comments have been addressed

Reviewer #3: (No Response)

Reviewer #4: (No Response)

2. Is the manuscript technically sound, and do the data support the conclusions?

Reviewer #1: (No Response)

Reviewer #3: Partly

Reviewer #4: Yes

3. Has the statistical analysis been performed appropriately and rigorously? 

Reviewer #1: (No Response)

Reviewer #3: No

Reviewer #4: Yes

4. Have the authors made all data underlying the findings in their manuscript fully available?

Reviewer #1: (No Response)

Reviewer #3: Yes

Reviewer #4: Yes

5. Is the manuscript presented in an intelligible fashion and written in standard English?

Reviewer #1: (No Response)

Reviewer #3: Yes

Reviewer #4: Yes

6. Review Comments to the Author

Reviewer #1: (No Response)

Reviewer #3: From the data presented and the statistical input that was implemented the investigators have concluded that. an increase in the number of ADR reports was noted in the intervention group during the intervention period, the changes in the "knowledge related to behavior” and in the "behavior related to reporting" score was significantly higher in the intervention group. Also specialist physicians and nurses, fulfilling additional positions and those working in other places demonstrated a high rate of report.

The data analysis approach is fairly routine and appears to be sufficient. Also the results are striking. However, the study lacks a rigorous study design justification. Three institutions were randomized, but the unit of analysis is the individual. There is no mention or investigation of the intraclass correlation in this type of setting and how that could impact on the varied analyses and the impact on the type I error. There is no central hypothesis. Other statistical issues include:

1. The multiple comparison testing issue is totally ignored, especially in Tables 3 and 4.

2. Tables 5 and 6 would suffice for a descriptive analysis. However, where is the institution effect in these tables?

3. There is insufficient tabular description of the number of cases , specialties, etc. within each of the institutions.

4. One of the key points mentioned by the investigators is “Education and training of medical staff in good pharmacovigilance practice is necessary for adequate ADR reporting and for the reduction of ADR rates and lower the costs in the healthcare system”. There is no analysis or meaningful discussion of costs considerations.

5. Reporting tools were conveniently telephone or online, but there is no validation of responses.

The investigators would do well to have this manuscript thoroughly edited by a statistician.

Reviewer #4: I really appreciate reading your work. I feel that the results are rather interesting. The employed methods are sound and the data are rich of info. I have just minor comments.

1. T-tests, as well as any other tests, must satisfy some assumptions. These assumptions are overlooked in the analysis. Please, provide evidence that the assumptions behind the t-test are fulfilled, to avoid misleading conclusions. Non-parametric tests could be used instead.

2. The use of regressions is sound. Nevertheless, to be confident on the obtained results, the authors must check for the Gauss-Markov assumptions. Inference on regression parameters is valid if and only if those assumptions are fulfilled. Moreover, I am wondering if multicollinearity may be an issue. Please, extend the regression model to include interactions. In general, the analysis of residuals may be included. It would be very useful to evaluate the reliability of the results.

7. PLOS authors have the option to publish the peer review history of their article (what does this mean?). If published, this will include your full peer review and any attached files.

Reviewer #1: No

Reviewer #3: No

Reviewer #4: No

---

## [Author Response · Author response to Decision Letter 1]

2 Apr 2020

To: Karen Cohen

Academic Editor 

PLOS ONE

Re: PONE-D-19-24600R1 “Increasing Adverse Drug Reaction Reporting – How can we do better?”

Dear Editor,

Thank you for your letter of March 2, 2020, and for the attached reviews. We have found your comments and the reviewers' comments most helpful and have addressed them in our revised version.

As for the comments:

Comments to the Author

1. If the authors have adequately addressed your comments raised in a previous round of review and you feel that this manuscript is now acceptable for publication, you may indicate that here to bypass the “Comments to the Author” section, enter your conflict of interest statement in the “Confidential to Editor” section, and submit your "Accept" recommendation.

Reviewer #1: All comments have been addressed

Reviewer #3: (No Response)

Reviewer #4: (No Response)

2. Is the manuscript technically sound, and do the data support the conclusions?

Reviewer #1: (No Response)

Reviewer #3: Partly

Reviewer #4: Yes

3. Has the statistical analysis been performed appropriately and rigorously? 

Reviewer #1: (No Response)

Reviewer #3: No

Reviewer #4: Yes

4. Have the authors made all data underlying the findings in their manuscript fully available?

Reviewer #1: (No Response)

Reviewer #3: Yes

Reviewer #4: Yes

5. Is the manuscript presented in an intelligible fashion and written in standard English?

Reviewer #1: (No Response)

Reviewer #3: Yes

Reviewer #4: Yes________________________________________

6. Review Comments to the Author

Reviewer #1: (No Response)

Reviewer #3: From the data presented and the statistical input that was implemented the investigators have concluded that. an increase in the number of ADR reports was noted in the intervention group during the intervention period, the changes in the "knowledge related to behavior” and in the "behavior related to reporting" score was significantly higher in the intervention group. Also specialist physicians and nurses, fulfilling additional positions and those working in other places demonstrated a high rate of report.

The data analysis approach is fairly routine and appears to be sufficient. Also the results are striking. However, the study lacks a rigorous study design justification. Three institutions were randomized, but the unit of analysis is the individual. There is no mention or investigation of the intraclass correlation in this type of setting and how that could impact on the varied analyses and the impact on the type I error. There is no central hypothesis. 

Answer: 

The research groups were compared in terms of background characteristics and were found to be similar in most variables, so there is no reason to assume there is an Intraclass correlation. The main hypothesis: after the intervention program there would be more ADRs reporting among medical professionals (physicians and nurses) in the intervention group compared to control group and to ADRs reporting base line. This was added to the methods section (page 10, line 204)

Other statistical issues include:

1. The multiple comparison testing issue is totally ignored, especially in Tables 3 and 4.

Answer: 

The variables compared in Tables 3 and 4 include two groups, thus there is no need to correct for multiple comparisons. The p value of all the variables with 3 groups were found to be not significant so there is no need to correct for multiple comparison. 

2. Tables 5 and 6 would suffice for a descriptive analysis. However, where is the institution effect in these tables?

Answer: 

Indeed, beyond the intervention there could be also the influence of the medical institution itself. However, there were no significant differences between the participants from the various medical centers with regard to the basic demographic characteristics. 

This information was added to the results section page 11 line 236: "433 (81.5%) medical staff members, physicians and nurses, completed the questionnaire twice, before and after the intervention. 47.8% of the participants were from the "A" medical center, 28.4% from "B" and 23.8% from "C". Distribution by gender was 69.1% females and 30.9% males. 73% were nurses and 27% physicians. No selection bias was found between the staff members completed the questionnaire the first time and those who completed it twice. No differences in personal or professional variables were found between the intervention group ("A" medical center) and the control group ("B" and "C" medical centers), except for the ratio between physicians and nurses and the subjects country of origin and average age." 

3. There is insufficient tabular description of the number of cases, specialties, etc. within each of the institutions.

Answer:

This data was added to the results section of the revised manuscript, page 11 line 236. 

4. One of the key points mentioned by the investigators is “Education and training of medical staff in good pharmacovigilance practice is necessary for adequate ADR reporting and for the reduction of ADR rates and lower the costs in the healthcare system”. There is no analysis or meaningful discussion of costs considerations.

Answer:

Thanks for the comment. Since reduction of costs due to ADRs reporting was not the scope of this study, we removed this section from the revised manuscript. 

5. Reporting tools were conveniently telephone or online, but there is no validation of responses.

Answer:

This was one of the questions that was asked in the questionnaire. "ADRs were reported through the Ministry of Health computerized website (for both the intervention and the control group) or documented in binders available only in the departments of the intervention group." (Page 9, line 171). Therefore, it seems that there is no need to validate it. 

The investigators would do well to have this manuscript thoroughly edited by a statistician.

Answer:

 A statistician reviewed the revised manuscript, as suggested. 

Reviewer #4: I really appreciate reading your work. I feel that the results are rather interesting. The employed methods are sound and the data are rich of info. I have just minor comments.

1. T-tests, as well as any other tests, must satisfy some assumptions. These assumptions are overlooked in the analysis. Please, provide evidence that the assumptions behind the t-test are fulfilled, to avoid misleading conclusions. Non-parametric tests could be used instead. 

Answer:

Thank you for your remark. In the "Methods" section, there is a section of data analysis in page 10, line 209: "Means and standard deviation were used for continuous variables and examined by T or One-Way ANOVA/Mann Whitney tests based on the variables distribution." We added that: "The score of "Knowledge related to behavior" and the score of "behavior related to reporting" didn’t distribute normally, thus we used a- parametric test (Mann Whitney). Page 10 line 2011.

2. The use of regressions is sound. Nevertheless, to be confident on the obtained results, the authors must check for the Gauss-Markov assumptions. Inference on regression parameters is valid if and only if those assumptions are fulfilled. Moreover, I am wondering if multicollinearity may be an issue. Please, extend the regression model to include interactions. In general, the analysis of residuals may be included. It would be very useful to evaluate the reliability of the results.

Answer:

Thank you for your Suggestion. We have checked again for Gauss-Markov assumptions and the Inference on regression parameters was found valid and there was no multicollinearity and interactions between the independent variables. 

7. PLOS authors have the option to publish the peer review history of their article (what does this mean?). If published, this will include your full peer review and any attached files.

Do you want your identity to be public for this peer review? For information about this choice, including consent withdrawal, please see our Privacy Policy.

Reviewer #1: No

Reviewer #3: No

Reviewer #4: No

Sincerely,

Miri Potlog shchory

---

## [Editor Report · Decision Letter 2]

12 May 2020

PONE-D-19-24600R2

Increasing Adverse Drug Reaction Reporting - How can we do better?

PLOS ONE

Dear Dr. Shchory Potlog,

Thank you for submitting your manuscript to PLOS ONE. After careful consideration, we feel that it has merit but does not fully meet PLOS ONE’s publication criteria as it currently stands. Therefore, we invite you to submit a revised version of the manuscript that addresses the points raised during the review process.

It is essential that the following points be addressed:

Please can you provide some information as to how the 3 sites were allocated to either "intervention" or "control"

The reviewer's comment re site-specific factors has not been addressed. There may be differences between the sites that contribute to the differences seen, that are not captured in the variables presented in the manuscript. Please address this comment more comprehensively, and include this in the limitations

 Please include description of the model building strategy for multivariable models. How were the included variables chosen for the multivariable models presented? Are there any other variables included in these models? If yes please specify.

The discussion and conclusion does not adequately address the fact that the impact of the intervention waned with time. It is therefore by no means certain that this intervention would improve safety of medicine use in the long term. There is little basis for reaching the conclusion that this intervention would lead to "overall reduced health costs"- suggest remove this statement from the conclusion.

Please add a paragraph on the limitations of this study. This should include the small number of sites included.

We would appreciate receiving your revised manuscript by  30 May 2020. To enhance the reproducibility of your results, we recommend that if applicable you deposit your laboratory protocols in protocols.io, where a protocol can be assigned its own identifier (DOI) such that it can be cited independently in the future. For instructions see: http://journals.plos.org/plosone/s/submission-guidelines#loc-laboratory-protocols

We look forward to receiving your revised manuscript.

Kind regards,

Karen Cohen

Academic Editor

PLOS ONE

---

## [Author Response · Author response to Decision Letter 2]

3 Jun 2020

To: Karen Cohen

Academic Editor PLOS ONE

Re: PONE-D-19-24600R2

Increasing Adverse Drug Reaction Reporting - How can we do better?

PLOS ONE

Dear Editor,

Thank you for your letter of May 12, 2020, and for the comments. We have found your comments most helpful and have addressed them in our revised version.

As for the comments:

"It is essential that the following points be addressed:

- Please can you provide some information as to how the 3 sites were allocated to either "intervention" or "control" The reviewer's comment re site-specific factors has not been addressed. There may be differences between the sites that contribute to the differences seen, that are not captured in the variables presented in the manuscript. Please address this comment more comprehensively, and include this in the limitations"

Answer: 

Three medical centers have participated in the study (Center "A", "B" and "C"). "The randomization among the centers was raffled by an external person who was not related to the study." (Page 7, lines 131-132). "Medical center "A" was randomly selected to be the intervention group. Medical centers "B" and "C" were merged and served as the control group" (Page 7, lines 129-131). "The medical centers selected for the study were public hospitals serving an urban and rural population of 0.5-1 million people each." (Page 7, lines 123-124). The participants were from the: "internal medicine divisions from three public medical centers in Israel: "A", "B" and "C", while each division served as a cluster." (Page 7, lines 121-123). This information is presented in the revised manuscript, as suggested.

- "Please include description of the model building strategy for multivariable models. How were the included variables chosen for the multivariable models presented? Are there any other variables included in these models? If yes please specify."

Answer:

The following paragraph was added to the method section (page 11 lines 231-237): "The building strategy for multivariable models was forced all the independent variables to one block. Both statistical and clinical justifications were considered. The models included all the variables that were found to be significant (p<0.05) in the univariate data analysis, and covariates that were important to controlling for, as a baseline characteristics, according to the research questions. All the independent variables that were included in the analyses were presented in the results of the multivariable models." 

- "The discussion and conclusion does not adequately address the fact that the impact of the intervention waned with time. It is therefore by no means certain that this intervention would improve safety of medicine use in the long term. There is little basis for reaching the conclusion that this intervention would lead to "overall reduced health costs"- suggest remove this statement from the conclusion.

Please add a paragraph on the limitations of this study. This should include the small number of sites included."

Answer:

It is mentioned in the discussion section that: "The trend presented in the data that the rate of reports decreased over time after the intervention program was discontinued implies that continued intervention may be required to maintain the high rate of reports." (Page 20 lines 377-380) and also that: "However, as distance from the intervention period increased, the amount of reports decreased eventually reverting to the trend that was observed at the baseline. (Page 21 lines 384-386) 

We also added that: "The impact of the intervention waned with time. An example to this trend was shown in an educational intervention program to improve physicians' reporting of ADRs. In this study the reporting rate in the intervention group increased during the intervention, while it gradually decreased through 13 months of follow-up. [9]." (Page 21 lines 386-390) 

"Due to the observation that the reporting rate decreased with time upon the finalization of the intervention period it seems that….." This was added to the conclusions section (Page 24, lines 458-460)

We agree that: "It is therefore by no means certain that this intervention would improve safety of medicine use in the long term and also reduce health costs". As suggested, this statement was removed from the conclusion.

We added a paragraph on the limitations of the study, as suggested. (Pages 24-25 Limitations 

The study was carried out in internal divisions of only 3 public medical centers, and therefore may not have an external validation in other hospitals. 

The study included only physicians and nurses, therefore the results cannot be applied to additional medical professionals. 

There were some differences in the basic characteristics between the clusters (hospitals) which may have affected the quality of the intervention to a certain extent. However, after adjusting for the demographic variables, we can assume that the results of the study are indeed due to the effect of the intervention. 

We cannot assume from this study that the effect of the intervention would improve safety of medicine use in the long term and reduce health costs.

---

## [Editor Report · Decision Letter 3]

19 Jun 2020

Increasing Adverse Drug Reaction Reporting - How can we do better?

PONE-D-19-24600R3

Dear Dr. Shchory Potlog,

We’re pleased to inform you that your manuscript has been judged scientifically suitable for publication and will be formally accepted for publication once it meets all outstanding technical requirements.

Kind regards,

Karen Cohen

Academic Editor

PLOS ONE
---

## [Editor Report · Acceptance letter]

27 Jul 2020

PONE-D-19-24600R3 

Increasing Adverse Drug Reaction Reporting - How can we do better? 

Dear Dr. Potlog Shchory:

I'm pleased to inform you that your manuscript has been deemed suitable for publication in PLOS ONE. Congratulations! Your manuscript is now with our production department. 

Kind regards, 

on behalf of

Dr. Karen Cohen 

Academic Editor

PLOS ONE